# Using Hormone Data and Age to Pinpoint Cycle Day within the Menstrual Cycle

**DOI:** 10.3390/medicina59071348

**Published:** 2023-07-23

**Authors:** Elinor Hills, Mark B. Woodland, Aparna Divaraniya

**Affiliations:** 1Oova, Inc., 335 Madison Avenue, New York, NY 10017, USA; 2Department of Obstetrics and Gynecology, Drexel University College of Medicine, Philadelphia, PA 19129, USA; 3Department of Obstetrics and Gynecology, Reading Hospital-/Tower Health, West Reading, PA 19611, USA; 4Pennsylvania State Board of Medicine, Harrisburg, PA 17110, USA

**Keywords:** ovulation tracking, quantitative monitoring of the menstrual cycle, cycle mapping, luteinizing hormone, fertility tracking

## Abstract

*Background and Objectives*: Menstrual cycle tracking is essential for reproductive health and overall well-being. However, there is still an over-reliance on estimations that standard cycles are 28 days long, divided evenly between the follicular and luteal phases. Due to the variability of cycle length and cycle phase lengths, common methods of identifying where an individual is in their cycle are often inaccurate. This study used daily hormone monitoring obtained through a remote hormone-monitoring platform to evaluate hormone levels across a menstrual cycle to identify nuances in the follicular and luteal phases in individuals of different age groups. *Materials and Methods*: This study used a remote fertility testing system that quantitatively tracks luteinizing hormone (LH) and pregnanediol-3-glucuronide (PdG) through urine tests read by an AI-powered smartphone app. The study analyzed cycle data from 1233 users with a total of 4123 evaluated cycles. Daily levels for LH and PdG were monitored across multiple cycles. *Results*: This study determined that calculated cycle lengths tended to be shorter than user-reported cycle lengths. Significant differences were observed in cycle phase lengths between age groups, indicating that follicular phase length declines with age while luteal phase length increases. Finally, the study found that if an individual’s age, first cycle day, and current hormone levels are known, population-level hormone data can be used to pinpoint which cycle phase and cycle day they are in with 95% confidence. *Conclusions*: At-home hormone monitoring technologies can allow patients and clinicians to track their cycles with greater precision than when relying on textbook estimations. The study’s findings have implications for fertility planning, clinical management, and general health monitoring. Prior to this study, no standard existed for pinpointing where a person was in their cycle through only one measure of LH and PdG. These findings have the potential to fill significant gaps within reproductive healthcare and beyond.

## 1. Introduction

The menstrual cycle involves the production of hormones essential to regulating a myriad of processes within the body, including follicular growth, ovulation, and the development of the endometrium [1,2,3]. Additionally, research has expanded on how fertility hormones can affect individuals beyond their reproductive health, such as sleep hygiene [4,5], premenstrual syndrome (PMS) and premenstrual dysphoric disorder (PMDD) management [6,7], diet and nutrition [8,9,10], mental health [11], and athletic performance [12,13].

The menstrual cycle is divided into two primary ovarian phases: the follicular phase and the luteal phase. The development of the ovarian follicle characterizes the follicular phase and culminates in ovulation, while the luteal phase is characterized by the post-ovulation formation and maintenance of the corpus luteum, which produces progesterone to prepare the uterus for a potential pregnancy. Accurate knowledge of menstrual cycle length and phase lengths is important for individuals seeking to manage their fertility. These factors play a critical role in predicting ovulation and fertile windows [3,6].

The benefits of cycle tracking are well-established. However, in clinical and at-home settings, pinpointing ovulation and the duration of the cycle and its phases still often rely on outdated standards and estimations [14,15,16]. Research has highlighted the need to move away from the clinical practice of comparing fertility hormone levels to the mean profile [17].

Research has found that fewer than 13% of menstruating individuals can correctly identify when they are ovulating [18]. This is likely due to two factors. First, it has long been believed that a healthy menstrual cycle lasts for exactly 28 days. Second, there is a prevailing assumption that the cycle is split evenly between the follicular and luteal phases, with ovulation occurring in the middle, on cycle day (CD) 14 [15,19].

However, in recent years, studies have found that most menstrual cycles do not adhere to these parameters [20,21]. In fact, only a small fraction of individuals ovulate on CD14, even among those with regular menstrual cycles [20]. Despite this, many popular ovulation predictor kits and period-tracking apps still rely on outdated assumptions, which often result in inaccurate predictions of the fertile window.

With the emergence of digital health technologies, accurate at-home hormone tracking has become more accessible and convenient [22,23]. Oova is a testing system that allows users to track their luteinizing hormone and pregnanediol-3-glucuronide (PdG) fluctuations throughout their cycles. The Oova platform collects a combination of self-reported data and quantitative hormone measurements through urine to identify users’ unique fertile windows and predict ovulation [18,24].

Many at-home fertility monitoring and testing kits focus on identifying the fertile window and confirming ovulation [25,26], but the full ability of these tools to map the hormone cycle with more granularity has been largely unexplored.

A longitudinal study was conducted using data from the Oova platform. We investigated how Oova’s cycle and phase length calculations compare to the textbook standards that are relied on by patients and in clinical practice. We also evaluated how cycle lengths and cycle phase durations differ across age groups to determine if at-home monitoring could provide accurate and precise information about an individual’s cycle.

Finally, we evaluated whether we could use population-level hormone data to pinpoint the phase of an individual’s cycle by knowing their age and hormone measurements at any given time. We hypothesized that Oova’s testing system could provide deeper insight into cycle phase length variability and hormone-level trends than textbook standards for cycle length calculation.

## 2. Materials and Methods

### 2.1. Hormone Monitoring

Oova is an at-home fertility testing system that quantitatively tracks LH and PdG through urine test cartridges. Urine samples are collected either in midstream or dip format and then scanned and interpreted by an AI-powered smartphone app. The kit is a non-invasive alternative to blood work that enables users to track their fertility hormones remotely. Users are guided through taking their first hormone scan when they start using the Oova app.

Oova’s test cartridges use advanced nanotechnology that adjusts for pH, normalizes hydration levels, and filters out non-specific binding. Urine can be collected in either midstream or dip test format. Test results are presented via lateral flow immunoassay. The Oova platform utilizes innovative computer vision algorithms to adjust for effects from lighting, shadows, and movement to ensure a good image is captured for analysis. Oova’s machine learning algorithms report each user’s unique hormone baseline levels. Fluctuations in daily levels are thus compared to the user’s personalized baseline instead of a pre-estimated cycle length. Previous verification studies have determined that Oova’s testing system quantitatively measures LH and PdG in urine, to a comparable degree to the ELISA quantified antigen standards and the AXXIN AX-2X-S reader device. The Oova device has undergone extensive research and development testing. Verification studies included lot-to-lot variation, limit of blank detection, and limit of quantitation calibration. All precision testing measures were conducted following the Clinical and Laboratory Standards Institute (CLSI) document EP05-A2’s definition of study protocol. These data are not yet published, but are currently being submitted for peer review. Additional studies comparing Oova’s results to serum LH and PdG results are currently underway to further explore Oova’s validity.

### 2.2. Study Design

Users who tracked at least one menstrual cycle on the platform were included in the study. Hormone cycle data was collected from Oova users between 20 April 2020 and 25 January 2023. The dataset was cleaned to exclude any incomplete entries. Data on age, height, weight, and period flow were self-reported and collected during participant onboarding and manually updated. Users navigated the Oova platform as guided by the algorithm.

### 2.3. Defining Ovulation and Cycle Length

Oova is a quantitative test and does not rely on LH levels to reach a certain threshold in order to detect a user’s LH peak. Oova determines the user’s LH peak by waiting for LH levels to rise above their established baseline levels to identify the fertile window. Oova’s device then confirms ovulation by detecting a rise in progesterone within 72 h after the highest LH levels are detected [27]. Only cycles for which an LH peak was recorded were included in this study. This methodology relies on the widely accepted assumption that LH and PdG levels are at their baseline prior to an LH surge [28,29,30].

The follicular phase was defined as the first day after a user stopped reporting bleeding to the date of the peak LH level. The luteal phase was defined as the days from the first day after ovulation to the day before the next menstrual cycle.

Average cycle lengths were initially self-reported into the Oova platform upon registration. When Oova users register for an account, they must enter their average cycle length. After the first tracked cycle, the cycle length calculation is automatically updated in the platform based on when a user logs subsequent periods. Cycles for which a user did not log menstruation were excluded from the study. For users with multiple tracked cycles, their average observed cycle length was calculated and used for analyses. For this study, CD1 was defined as the first day of menstruation. The platform registers the first day of menstruation if a user reports either one day of medium or heavy bleeding or two consecutive days of light bleeding. The Last Cycle Day (LCD) was defined as the last day of the luteal phase before CD1 of the next cycle.

### 2.4. Statistical Analysis

All analysis was conducted using R Version 4.2.2. Associated code and raw data will be available upon request.

## 3. Results

### 3.1. Result 1: Patient Demographic

A total of 1233 Oova users were included in this study, and a total of 4123 cycles were evaluated, as presented in Appendix A. Of the included users, 44.2% were in the 30–34 age group.

A majority of users included in this study self-identified as white, at 77.9% (8.2% self-identified as Asian, 4.2% as Black or African American, and 9.5% as another race(s)). “Other” includes users who self-identified as “More than one race”, “Native Hawaiian or Other Pacific Islander”, “American Indian or Alaska Native”, or “Unknown”. Users could also choose not to identify. No significant cofactors were identified across age groups.

### 3.2. Result 2: Calculated Cycle Lengths Tend to Be Shorter Than Self-Reported Cycle Lengths

We compared the self-reported cycle lengths to the calculated cycle lengths across age groups using a two-sided t-test to evaluate if there was a significant difference between the two measures.

Self-reported cycle lengths ranged from 19 to 36 days. Out of the total, 28.74% of users had a self-reported cycle length of <28 days, 47.59% had a self-reported cycle length of >28 days, and 23.67% had a self-reported cycle length of 28 days.

Calculated cycle lengths ranged from 9 to 101 days. Of the total included cycles, 73.65% had a length of <28 days, 21.12% had a cycle length of >28 days, and 5.23% had a cycle length of 28 days.

Overall, there was a significant difference in calculated versus self-reported cycle lengths across all age groups. Additionally, as indicated in Figure 1, our results demonstrate that calculated average cycle lengths were shorter than self-reported average cycle lengths across all age groups. A full table of significance levels can be seen in Appendix A.

This finding supports our hypothesis and existing research that individuals are not able to estimate their cycle length nor identify their fertile window with accuracy [6,16,18]. This finding further justifies the need for accessible testing methods in order to help individuals accurately interpret their cycles.

### 3.3. Result 3: Lengths of the Follicular Phase and Luteal Phase Decrease and Increase, Respectively, across Age Groups

The follicular phase and luteal phase lengths were calculated across each age group. As can be seen in Figure 2, significant differences were observed in the follicular phase lengths between the 30–34 and 35–39 age groups (*p* < 8.6 × 10^−5^) and between the 35–39 and 40–44 age groups (*p* < 1.7 × 10^−2^). These results indicate that the follicular phase duration changes over an individual’s lifespan, becoming shorter with age.

A significant difference was observed in the luteal phase lengths between the 25–29 and 35–39 age groups (*p* < 2.6 × 10^−4^) and between the 30–34 and 35–39 age groups (*p* < 1.5 × 10^−2^). These findings support that the luteal phase tends to get longer as the individual ages. Our data also highlight that the follicular and luteal phases vary in length, with the follicular phase being longer than the luteal phase, especially in individuals under the age of 35 years. This finding points to interpersonal variability in the duration of both the luteal and follicular phases. This challenges the common assertion that the luteal and follicular phases are the same length and bisect the menstrual cycle equally.

Furthermore, other studies have observed the opposite trend in age group data and cycle phase length [20,31]. While this study demonstrates that both the follicular and luteal phases become shorter with age, other studies have reported that the luteal phase stays constant over the lifespan [20,32,33].

Overall, a significant difference was observed between the variation in follicular and luteal phase lengths across all age groups. Specific significance levels can be viewed in Appendix A.

### 3.4. Result 4: Hormone Pattern Variability in the Follicular and Luteal Phases

Hormone patterns in the follicular and luteal phases were examined (Figure 3). Our findings suggest that while variability exists in cycle lengths and cycle phase lengths throughout the menstrual cycle, population-level trends in LH and PdG levels become apparent when looking at a large dataset.

Figure 3A–D depict the LH levels of users across different age groups during specific cycle days in both the follicular (A) and luteal (B) phases. Our results show that each age group’s LH levels across cycle days are relatively consistent. The variation observed in each age group is minimal, as can be observed with the tight error bars across cycle days in both the follicular and luteal phases. Panels (C) and (D) depict the PdG levels of users across different age groups during specific cycle days of the follicular and luteal phases, respectively. For PdG, we observed that levels remain low in the early follicular phase before starting to increase in the later cycle days of the follicular phase. In the luteal phase, PdG levels continue to increase before plateauing towards the middle/end of the phase.

Similar patterns are depicted across age groups, though the standard error is greater in the 25–29 age group during the follicular phase and in the 40–44 age group during the luteal phase.

This result illustrates how large datasets like Oova’s can reveal population-level hormone trends. Analyses of these datasets can be instrumental in efforts to optimize clinical practice and establish care guidelines.

### 3.5. Result 5: Hormone Patterns Enable Identification of the User’s Particular Cycle Day Based on Age

This result demonstrates that a user’s current menstrual phase and cycle day can be determined with a 95% confidence level by assessing their daily LH and PdG hormone levels alongside their age. Table 1 depicts the average (±SE) hormone levels for LH and PdG on each cycle day for the age group 30–34.

A summary of hormonal levels across the other age groups can be found in Appendix A. The days that repeat indicate an overlap between cycle days labeled as follicular and luteal across different users. This variability is expected as our earlier findings demonstrate that cycle phase lengths often do not evenly split the menstrual cycle.

These findings demonstrate how, by understanding the trends in hormone concentration changes throughout the cycle and over the lifespan, we can gauge with a 95% degree of confidence where a particular Oova user is in their cycle.

## 4. Discussion

Menstrual cycle tracking can benefit many individuals, including, but not limited to, people trying to conceive, those trying to avoid conception, or individuals managing a chronic reproductive condition like polycystic ovary syndrome (PCOS) or premenstrual dysphoric disorder (PMDD). Cycle tracking is most discussed in the context of fertility and is utilized by individuals trying to conceive or navigating infertility. Cycle length or phase duration changes may signal underlying health problems [34,35]. Age, body weight, and hormonal imbalances can impact menstrual cycle length and phase duration [36].

This study supports existing research that states that most individuals are not able to accurately estimate their cycle length, nor identify when they are fertile [18,37]. We observed that self-reported and calculated cycle lengths differ because individuals believe they have a 28-day cycle and are not aware of the variations in their bodies that occur from cycle to cycle.

This research also supports existing findings that have demonstrated greater variability across cycle lengths and cycle phase durations than historical definitions of the menstrual cycle would indicate [18,19,20,21]. By tracking real-time hormone data throughout the cycle on a large scale, we observe that cycle phase lengths vary across individuals and may change over the course of the lifespan.

It is well established that fertility in menstruating people declines with age [38], with most research citing a decrease in egg quantity and egg quality [38,39,40,41]. Less focus has been applied to age-related hormone changes across the menstrual cycle, possibly contributing to an individual’s ability to conceive.

This study highlights that interpersonal variability in the duration of both phases and the duration of each phase may shift intrapersonally over the course of the lifespan. This challenges the common assumption that the follicular and ovulatory phases are of equal length and split the menstrual cycle in half.

Studies across age groups have observed analogous patterns to this study in changes in follicular phase length [20,31]. Our study indicates that the follicular phase shortens over the lifespan; however, other studies found conflicting results about the changes, or lack thereof, in the duration of the luteal phase. While our data indicate that luteal phase lengths increased between the 25–29 and 35–39 age groups and between the 30–34 and 35–39 age groups, other studies have suggested that luteal phase duration does not change over the lifespan [20,32,33].

By understanding how hormone level trends change with age and concentrations of hormone levels vary over a cycle, we can determine where a user is in their cycle with a 95% confidence level. Oova’s large dataset can be utilized to reveal population-level trends in hormone levels at different points in the cycle. This information can inform clinical practice and care guidelines.

Innovations in fertility technology, such as Oova, have the potential to improve the standard of care for fertility management and treatment by making high-sensitivity hormone testing more accessible and less reliant on invasive or qualitative measures. Current practice often relies on blood draws, ovulation predictor kits (OPKs), or basal body temperature (BBT) tracking to monitor the cycle. Each method has its shortcomings [16,24,42,43,44,45].

### 4.1. Implications of Findings beyond Tracking Fertility

The ability to accurately determine where an individual is in their menstrual cycle has several key implications, both in the context of fertility and beyond. Understanding where a patient is in their cycle can help clinicians optimize the timing of medication administration, namely in conditions such as polycystic ovary syndrome (PCOS) and premenstrual dysphoric disorder (PMDD) [46]. This knowledge can support individuals in navigating self-management of their cycle-related symptoms. Because there is a high degree of variability in experiences of cycle-related symptoms tracking [47,48], treatment must be personalized and individualized rather than based on cross-population estimations or averages [6,46].

In such cases, at-home hormone monitoring technologies with high sensitivity, such as Oova, could be useful for accurately identifying where an individual is in their cycle beyond just predicting and confirming ovulation. By providing real-time hormone level data, these devices can help individuals, and their providers, identify patterns and monitor their response to treatment or lifestyle modifications.

For doctors and patients alike, much of the ambiguity around the menstrual cycle stems from an outdated and often inapplicable definition of a “normal” cycle [15,19]. Our results support existing hypotheses about gaps in knowledge around cycle length. For patients, an inability to define and identify cycle phases can have implications for managing fertility, chronic condition treatment, and general health and well-being. This insufficiency manifests in other far-reaching manners for providers and medical researchers.

The medical research community lacks a consistent means for operationalizing the cycle [16,24,42,43,44,45]. This need is present in studies where menstruation is the core interest and those requiring controlling for hormone changes to accurately investigate another variable. Researchers have called attention to the issue in recent years to emphasize the need for a standard definition and method for categorizing menstrual cycle phases [16]. Our study provides guidelines for understanding cycle phases and hormone concentrations throughout the cycle. This provides an alternative to relying heavily on estimations based on cycle length averages. The presented guidelines have the potential to improve study design and research practice.

### 4.2. Further Directions

This study’s findings will be applied to create similar models for individuals with various reproductive health disorders, such as PCOS and PMDD. Our goal is to utilize this research to help clinicians and patients better understand the nuances and variability in hormone profiles across cycle days regardless of where they are in their reproductive journey and despite diagnoses that could cause irregularities. Not only would this have implications for the diagnostic process, but it could also inform treatment protocol and help track a patient’s response to treatment.

This study’s findings also suggest a potential extension for this testing technology to aid pregnancy monitoring. While the test is not developed to confirm pregnancy, Figure 3 indicates a pattern at the end of the luteal phase when some users’ LH levels begin to rise post-ovulation while the PdG levels continue to rise. This pattern can be a sign of early pregnancy. Further research will explore the potential for using the Oova platform as an early pregnancy indicator by developing higher thresholds for measuring relevant biomarkers.

### 4.3. Limitations

The study population was self-selecting, consisting of people wanting to track ovulation to try to conceive. Some participants may have also used the Oova platform because they had difficulty tracking ovulation with different tools or means. This suggests that users within our cohort had irregular cycles or ovulatory patterns that fell outside the “normal” range, which may have influenced our findings.

Furthermore, because we decided to exclude cycle days for which we had fewer than 100 samples, our analysis of the 25–29-year-old age group and the 40–44-year-old age group were limited. Initial findings suggest that mean hormone levels of the 25–29-year-old group were lower than the means for older age groups. This finding is contrary to what would be expected among the younger demographic and warrants additional analysis with a larger sample size.

The inability to ensure adherence to testing guidelines was also a limitation, as was the potential for user error. Users opted to use the platform on their own or at their doctor’s suggestion. Users did not go through a training process and only received the standard onboarding instructions presented to them within the app. If discrepancies in user adherence were present, the large sample size would mitigate the effect of user error on our overall findings.

Users may stop testing once they see that they have ovulated to save test strips, which can limit the accuracy of cycle tracking. Reliance on user input of estimated cycle length for the first cycle, as well as the user entering information about their period correctly, was a limitation. However, the analysis found that calculated cycle lengths were more accurate than estimates.

## 5. Conclusions

At-home hormone monitoring technologies like Oova’s testing system provide accurate, quantitative hormone readings that can be used to improve fertility guidance, clinical treatment, and health management. This study’s findings establish that population-level hormone data can allow us to pinpoint where an individual is in their cycle with only knowledge of their age, the first day of their cycle, and hormone measurements from a single point in time.

## Figures and Tables

**Figure 1 medicina-59-01348-f001:**
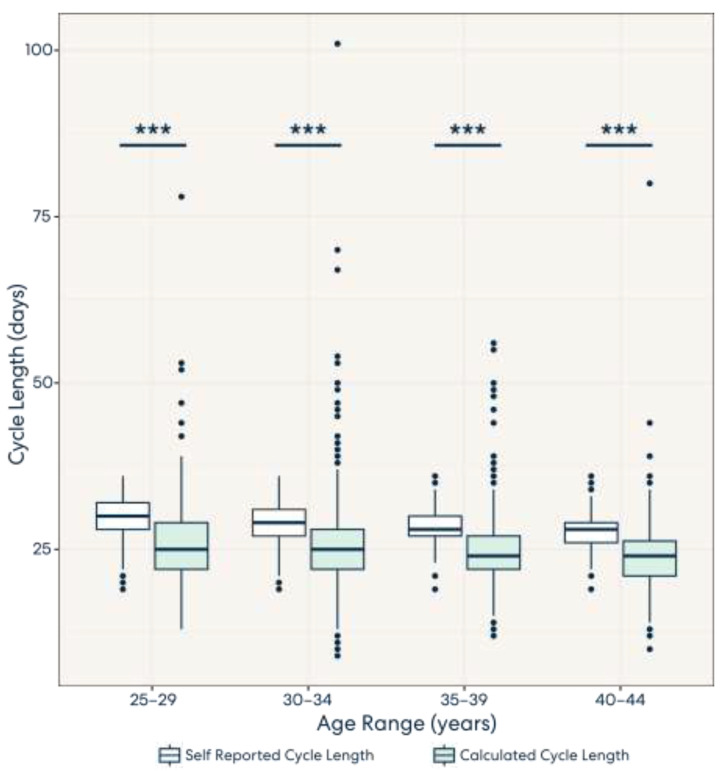
Self-reported and calculated cycle lengths across age groups. Across all age groups, calculated cycle lengths tended to be shorter than self-reported cycle lengths, with a significance level of *** *p* < 0.001.

**Figure 2 medicina-59-01348-f002:**
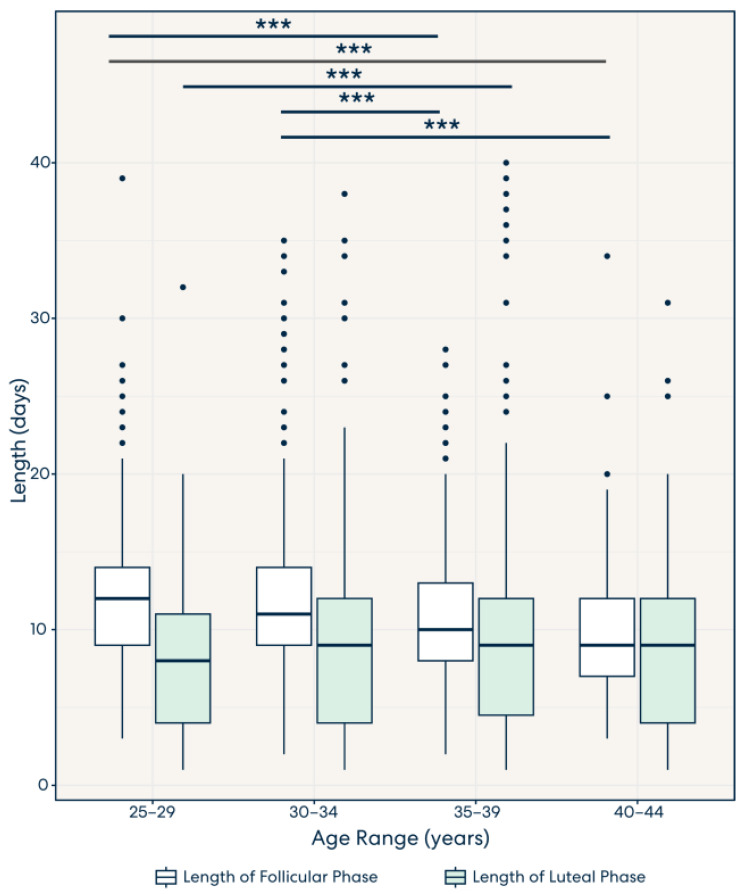
The distribution of the follicular and luteal phases across all age groups, with a significance level of *** *p* < 0. 001. Significance was found in follicular phase lengths between 25–29 and 30–34, and 35–39 and 40–44, but not displayed in this graph. Significance was also found in luteal phase lengths between 25–29 and 30–34, 25–29 and 40–44, and 30–34 and 35–39, but are not displayed in this graph. For a full list of associated *p*-values, see Appendix A.

**Figure 3 medicina-59-01348-f003:**
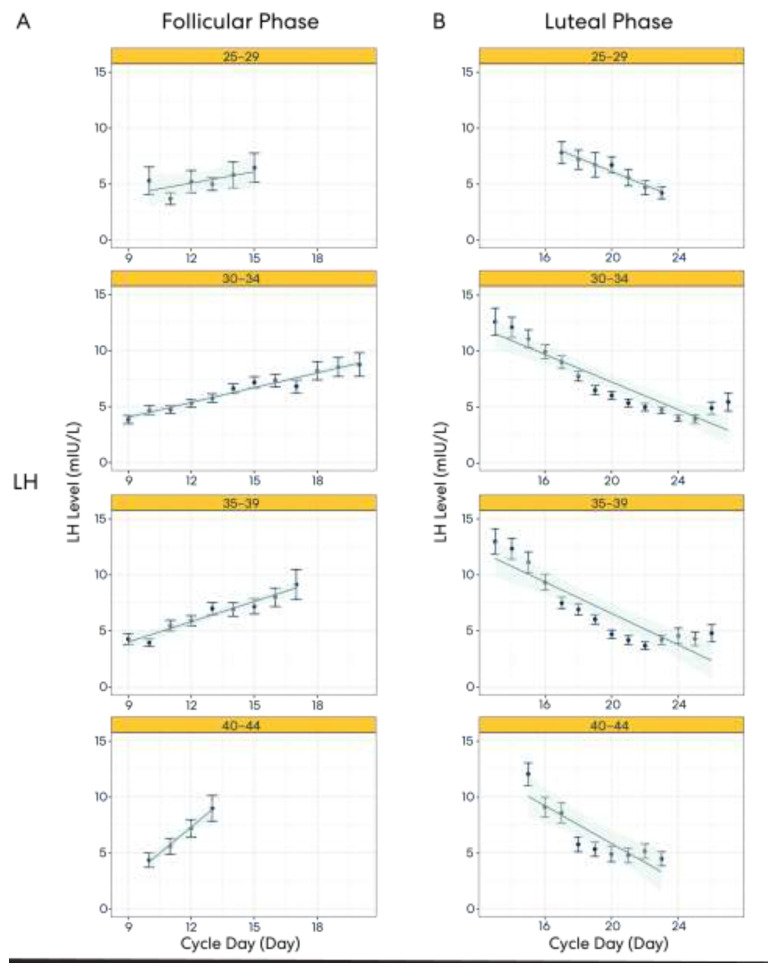
Panels (**A**,**B**) depict LH levels across age groups in follicular and luteal phases. Panels (**C**,**D**) depict PdG levels across age groups in follicular and luteal phases. Each panel shows the LH or PdG data for the specific phase for the designated age group. The data points indicate the mean hormone level on a specific cycle day. The error bars indicate the standard error. The shaded region designates the confidence interval. LH levels are shown in mIU/L while PdG is in ng/mL.

**Table 1 medicina-59-01348-t001:** The average (±SE) LH and PdG levels for each day of the cycle for the 30–34 age group. N represents the number of individuals included in the calculation of the average hormone levels for each cycle day for that phase.

	Follicular			Luteal		
Cycle Day	N	LH	PdG	N	LH	PdG
9	255	3.83 ± 0.36	1.95 ± 0.20			
10	405	4.64 ± 0.41	2.03 ± 0.15			
11	485	4.72 ± 0.33	2.08 ± 0.12			
12	514	5.30 ± 0.35	2.15 ± 0.13			
13	489	5.74 ± 0.39	2.11 ± 0.60	115	12.57 ± 1.22	5.48 ± 0.60
14	459	6.59 ± 0.44	2.11 ± 0.14	194	12.06 ± 0.91	5.57 ± 0.41
15	378	7.15 ± 0.50	2.33 ± 0.15	264	11.05 ± 0.79	6.11 ± 0.36
16	301	7.29 ± 0.57	2.84 ± 0.24	359	9.89 ± 0.62	6.37 ± 0.32
17	247	6.79 ± 0.57	2.94 ± 0.30	433	8.95 ± 0.57	8.19 ± 0.34
18	211	8.19 ± 0.82	3.11 ± 0.33	477	7.71 ± 0.45	8.63 ± 0.32
19	157	8.52 ± 0.85	3.32 ± 0.39	504	6.44 ± 0.41	9.92 ± 0.33
20	109	8.72 ± 1.04	4.10 ± 0.55	515	5.98 ± 0.37	10.23 ± 0.33
21				533	5.32 ± 0.39	10.44 ± 0.32
22				518	4.94 ± 0.35	11.12 ± 0.33
23				479	4.67 ± 0.29	11.78 ± 0.34
24				361	3.96 ± 0.30	11.45 ± 0.39
25				248	3.85 ± 0.38	11.79 ± 0.48
26				183	4.85 ± 0.55	11.02 ± 0.55
27				126	5.40 ± 0.81	11.36 ± 0.70

## Data Availability

Data regarding any of the subjects in the study has not been previously published unless specified. Data will be made available to the editors of the journal for review or query upon request. The data presented in this study are available on request from the corresponding author.

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
