# Peer review of "Using Hormone Data and Age to Pinpoint Cycle Day within the Menstrual Cycle"

_medicina, 2023, doi:10.3390/medicina59071348_

Round 1
Reviewer 1 Report
Reviewer comments on “Unlocking the mysteries of menstrual cycles: Using Hormone data and age to predict cycle day with precision” by Hills et al
First, I see this study as novel, extremely informative, and a foreshadow of the numerous insights we will gain regarding the reproductive cycle as quantitative at home monitoring systems become more common. That women inaccurately understand their average cycle length is not new news but the idea that population-based means may help convert an individual hormone value into a more accurate cycle day prediction using population means is novel. I look forward to additional new insights as this technology moves forward.
Minor edits required:
1. Line 102 states the correlation of the urinary hormone testing system used to blood levels but does not provide a reference. Please provide one, even if its unpublished data.
2. Line 308 in discussion references figure 6 but I think this is a typo and refers to figure 3.
3. The supplementary data are extremely useful, and I applaud the authors for providing them. I do see missing daily hormone means for cycle day 16 in sup table 5 and cycle day 14 in sup table 13. Please add these for a more complete picture.
4. Figure 3 when printed at normal size is very difficult to read, especially the shaded area representing confidence intervals (a critical set of data from this work). Perhaps separating into 2 figures (or 3.a and 3.b) that each take a page will help readers. I needed to increase the online size to 300% to see data clearly.
Other minor suggestions:
1. Legends for figs 1&2 have three ‘levels’ of p values, yet all the presented differences are p<0.001. Eliminating the irrelevant p values may clarify a bit.
Additional comments to consider that may improve an already very good paper:
It would be of interest to know how many of the cycles in the data set missed an LH peak. As I recall, data published using the qualitative Clearblue monitor in regular cycling women shows 8-10% of cycles “miss” the LH peak and these women are counselled to add a second daily test.
It is notable that the mean BMI for all but the 30–34-year-old groups is in the overweight category. It would be useful and perhaps informative if the daily hormone analysis was redone using ONLY women with normal BMI. Do the results change at all?
The mean hormone levels reported for 25-29 years olds are lower than the 30 somethings and even lower in some cases than 40+ year olds. This seems contrary to what one would expect of healthy ovulating women. I also note a lot of missing daily hormone data (211 women contributing only around 110 cycles of data per day). Further analysis of this and perhaps some comments in the discussion may be of benefit.
Reviewer 2 Report
Reviewer report is attached.

Reviewer 3 Report
I commend the authors for the article and I encourage them to continue with the revisions.
I will focus on two crucial elements. The title claims that the cycle day is predicted with precision. I have my doubts. First, they need to clearly state how ovulation was decided given that no ultrasound was performed. Once this is validated somehow, the figure 3 betrays the claim of precision. Multiple continuous cycle days, whether there are in the follicular and luteal phase, have overlapping confidence intervals. As a result, one can have 2, 3, 4 or more days with similar results in one given phase. I can see one could state that given a single day result one could predict that it is found early or late follicular phase or early or late luteal phase, but I don’t see the evidence that one can precisely ascribe it to a particular day of the cycle. I would also like to see a normogram of their date on each day such as the once found Clin Chem Lab Med 2015; 53(7): 1099–1108. I think this information would be very valuable. The rest of their findings are very interesting but they rest of the validity of the above. Keep up the good work.
Reviewer 4 Report
The data used seem interesting.
However, your analysis has severe limitations:
* you compare the average cycle length declared by the user at registration with the observed cycle length. These are two different kinds of information, and do not lead to the conclusion that women don't know their cycle length. It simply indicates that it varies from one cycle to the next, which is known. What's more, the statistic test you're using doesn't seem appropriate, as there are several cycles per woman.
* you propose to use the amplitudes of LH and PdG values without linking them either to ovulation or even to the LH peak: it can't be useful to use this type of reference, which is already available on the Internet, for example on Wikipedia. If reference amplitudes per day were to be used, it would have to be with a Day 0 that made sense in relation to ovulation.
The text itself is often more apologetic than is appropriate for an article describing results. It's only at the discussion stage that the results are commented on, not already in the results and their subheadings.
It seems that a major overhaul of the analysis would give you more success.
Round 2
Reviewer 4 Report
It does not seem appropriate to use a hormone assay within cycles to define cycle length. In fact, the study of cycle length is based simply on the recording of menstrual periods. You should delete this part of the paper.
The mean hormone value as a function of the time since the first day of the cycle and the 95% standard error of the mean are not a criterion for identifying the period of the cycle in question.
This is because :
1) the interval at 95% of the mean is a function of the number of cycles in the study
2) the length of the cycle varies from one cycle to another, as does the day of ovulation.
The average LH level on each day of the cycle since the start of the cycle cannot indicate whether we are before or after ovulation. Only progesterone can.
In short, it all boils down to comparing progesterone with a reference value for the pre-ovulatory and post-ovulatory periods. These values are available in the medical literature.
The pre-ovulatory period begins on the first day of the period, not the last.
You define the post-ovulatory period as beginning the day after ovulation, but have not defined your ovulation day.
You mention endometriosis as a context in which your method is useful. However, endometriosis does not vary the length of the cycle and treatments do not depend on the phase of the cycle.
You talk about "fertility in cisgender women" but there is no link between gender and the cycle.
The expression 'operationalizing the cycle' does not seem to be an English term.
You write 'Our study provides guidelines for defining cycle phase lengths': I don't see how this is true.
Round 3
Reviewer 4 Report
I apologize, but cannot accept the paper in its present form.
The article continues to fall far short on the three questions asked: only the second is adequate.
The first concerns the length of the cycle: hormone measurements are of no use in answering this question.
The third concerns the use of a single dose to determine whether the day is in the pre- or post-ovulatory: only progesterone can be used, LH cannot : values are frequently identical in pre and post-ovulatory.
The methods are also inadequate: notably, there is confusion between the amplitude of uncertainty of the mean and the amplitude of variation of individual values.
